# Low-Frequency Oscillations of In Vivo Ambient Extracellular Brain Serotonin

**DOI:** 10.3390/cells11101719

**Published:** 2022-05-23

**Authors:** Colby E. Witt, Sergio Mena, Lauren E. Honan, Lauren Batey, Victoria Salem, Yangguang Ou, Parastoo Hashemi

**Affiliations:** 1Department of Chemistry and Biochemistry, University of South Carolina, Columbia, SC 29208, USA; wittce@email.sc.edu (C.E.W.); honan@livemail.uthscsa.edu (L.E.H.); yangguang.ou@uvm.edu (Y.O.); 2Department of Bioengineering, Imperial College London, London SW7 2AZ, UK; sergio.mena19@imperial.ac.uk (S.M.); l.batey@imperial.ac.uk (L.B.); v.salem@imperial.ac.uk (V.S.); 3Department of Chemistry, University of Vermont, Burlington, VT 05405, USA

**Keywords:** serotonin, depression, fluctuations, serotonin transporter, antidepressants

## Abstract

Serotonin is an important neurotransmitter that plays a major role in many aspects of neuroscience. Fast-scan cyclic voltammetry measures fast in vivo serotonin dynamics using carbon fiber microelectrodes. More recently, fast-scan controlled-adsorption voltammetry (FSCAV) has been developed to measure slower, minute-to-minute changes in ambient extracellular serotonin. We have previously demonstrated that FSCAV measurements of basal serotonin levels give critical information regarding brain physiology and disease. In this work, we revealed the presence of low-periodicity fluctuations in serotonin levels in mouse hippocampi, measured in vivo with FSCAV. Using correlation analyses, we found robust evidence of oscillations in the basal serotonin levels, which had a period of 10 min and were not present in vitro. Under control conditions, the oscillations did not differ between male and female mice, nor do they differ between mice that underwent a chronic stress paradigm and those in the control group. After the acute administration of a selective serotonin reuptake inhibitor, we observed a shift in the frequency of the oscillations, leading us to hypothesize that the newly observed fluctuations were transporter regulated. Finally, we optimized the experimental parameters of the FSCAV to measure at a higher temporal resolution and found more pronounced shifts in the oscillation frequency, along with a decreased oscillation amplitude. We postulate that this work may serve as a potential bridge for studying serotonin/endocrine interactions that occur on the same time scale.

## 1. Introduction

Serotonin is a critical neurotransmitter that is of interest to many aspects of neuroscience. Several mechanisms of action regulate the extracellular concentration of serotonin, including synthesis (from 5-hydroxytryptophan), packaging, neuronal release (controlled by terminal and somatodendritic autoreceptors), reuptake mechanisms (*via* serotonin and other monoamine transporters) and metabolism [1,2,3].

Painting a full picture of in vivo serotonin chemistry at high temporal resolution is highly physiologically informative but is also challenging. We have utilized fast-scan cyclic voltammetry (FSCV) (frequency of 10 Hz) for a decade to measure the in vivo serotonin dynamics with carbon fiber microelectrodes (CFMs). Notable studies include work exploring different mechanisms of serotonin reuptake, serotonin dynamics in male and female mice, and a comparison of serotonin transmission between different brain regions [4,5,6,7,8]. From these prior works, we learned that serotonin is subject to profound regulatory mechanisms that limit its influence, likely due to the neurotoxicity associated with uncontrolled extracellular levels [9]. Recently, we applied fast-scan controlled-adsorption voltammetry (FSCAV) [10,11] to measure slower, minute-to-minute changes in ambient extracellular serotonin [12]. We found that ambient serotonin levels differed between brain regions and responded to selective serotonin transporter inhibitors (SSRIs) and inflammation [5,13].

In this work, we analyzed the dynamics of in vivo FSCAV data. We visually observed evidence of fluctuating chemical patterns in data collected from the mouse hippocampus. Using correlation analysis (autocorrelation and sliding window correlation (SWC)), we found a strong oscillation in basal serotonin levels with a period of approximately 10 min. We showed that these oscillations were most likely physiological because they were not present in the in vitro FSCAV recordings. We revealed via power spectrum density (PSD) analysis that, under control conditions, the oscillations did not differ between male and female mice. This finding was not surprising to us since, in the past, we found no differences in both evoked and basal serotonin between sexes [8]. We also found that the frequency of the hippocampal oscillations did not differ between control mice and mice that underwent a chronic stress paradigm, despite a substantial difference in the absolute levels between these cohorts [14]. After acute administration of SSRIs, we observed a shift in the frequency of the oscillations, which led us to theorize that slow serotonin fluctuations are controlled, in part, via transporters. Finally, we optimized various experimental parameters of FSCAV to measure at a higher temporal resolution, under the rationale that we may capture more information after SSRI administration. Indeed, when collecting data at a higher temporal resolution, we found larger shifts in the oscillation frequencies and captured other interesting features, such as a decreased oscillation amplitude. A possible explanation for these ambient oscillations is that they are connected to important endocrine fluctuations in the blood, which occur on the same temporal scale. This model may therefore serve as a critical bridge for studying serotonin–endocrine interactions, which are of critical, and timely, interest.

## 2. Materials and Methods

### 2.1. Chemicals and Reagents

Escitalopram oxalate (10 mg/kg, Sigma-Aldrich, St. Louis, MO, USA) was dissolved in sterile saline (0.9% NaCl solution, Hospira, Mountainside Medical Equipment, Marcy, NY, USA) and administered via intraperitoneal injection (*i.p.*) at 10 mg/kg and a volume of 5 mL/kg body weight. Urethane (Sigma-Aldrich, St. Louis, MO, USA) was dissolved in sterile saline at 25% *w*/*v* and administered at 7 µL/g mouse body weight for surgical anesthesia. Serotonin hydrochloride (Sigma-Aldrich, St. Louis, MO, USA) was dissolved in 1× phosphate buffered saline (Sigma-Aldrich, St. Louis, MO, USA) to obtain serotonin solutions for the in vitro calibrations. 

### 2.2. Animal and Surgical Procedure

The chronic unpredictable mild stress (CMS) paradigm occurred over a 16-week period based on previous models [15,16,17,18]. Two to three mild stressors (*e.g.*, food or water deprivation, confinement, and light during dark cycle) were performed a day before the neurochemical study. An extensive description of the protocol is given elsewhere [14]. 

Mice (C57BL/6J) (Jackson Laboratory, Bar Harbor, ME, USA) were injected with a 25% urethane solution based on body weight (7 µL/g). Following anesthetic administration, the mouse was placed into a stereotactic system (David Kopf Instruments, Tujunga, CA, USA) where body temperature was maintained via heating pad (Braintree Scientific, Braintree, MA, USA). Three holes were drilled into the skull of the mouse based off coordinates from the mouse brain atlas [19]. The working electrode was placed in the CA2 region of the hippocampus (CA2: −2.91, +3.35, −2.5), the stimulating electrode (insulated stainless-steel, diameter 0.2 mm, untwisted, Plastics One, Roanoke, VA, USA) was placed in the medial forebrain bundle (MFB: −1.58, +1.00, −4.80), and the pseudo-Ag|AgCl reference electrode was placed in the opposite hemisphere of the brain from the working and stimulating electrodes. Animal use followed NIH guidelines and complied with the University of South Carolina Institutional Animal Care and Use Committee under an approved protocol. 

### 2.3. Microelectrode Fabrication

Carbon fiber microelectrodes (CFMs) were made individually by aspirating a single carbon fiber (Goodfellow Corporation, Coraopolis, PA, USA) into a 0.6 mm × 0.4 mm glass capillary (A-M Systems, Inc., Sequim, WA, USA). The capillary was then pulled by a vertical puller (Narishige, Tokyo, Japan) to create a seal. The carbon fiber was then trimmed to 150 ± 3 µm. Liquion (LQ-1105, 5% by weight Nafion ^TM^, New Castle, DE, USA) was electrodeposited onto the surface of the carbon fiber by dipping and applying a constant potential of +1.0 V for 30 s. The electrode was then dried at 70 °C for 10 min and was used 24 h after the final product was crafted.

### 2.4. Hardware

FSCAV was performed using a Dagan potentiostat, (Dagan Corporation, Minneapolis, MN, USA), National Instruments multifunction device USB-6341 (National Instruments, Austin, TX, USA), WCCV 3.06 software (Knowmad Technologies LLC, Tucson, AZ, USA) and a Pine Research headstage (Pine Research Instrumentation, Durham, NC, USA). Data filtering (zero phase, Butterworth, 2 kHz lowpass) and signal smoothing were done within the WCCV software.

### 2.5. Experimental Procedures and Data Collection

FSCAV was performed in four steps: (1) the serotonin waveform (N-shaped waveform from 0.2 V to 1.0 V, to −0.1 V, and then back to 0.2 V, at 1000 V/s) [20], was applied at 100 Hz to minimize adsorption of serotonin to the electrode surface for 2 s; (2) a fixed potential (0.2 V) was applied for 10 s, allowing for maximized adsorption; (3) the serotonin waveform was then reapplied for 18 s at 10 Hz and the third cyclic voltammogram (CV) collected was used to measure ambient serotonin concentration; (4) a waiting period of 30 s was then applied until the next file was acquired. The duration of each of these steps was also changed to increase the acquisition frequency of FSCAV (see Results). The CVs were presented as charge vs. time and the oxidation peak for serotonin was integrated via custom software [21]. This current vs. time area (charge) was converted to concentration by using a linear calibration curve (when available, using four in vitro serotonin solutions of 10, 25, 50, and 100 nM serotonin) or, when calibrations were not available, by using a newly developed standardized neural network regression model trained with a large in vitro dataset (*vide infra*) [21]. Sensitivity for measuring basal serotonin, before and after changing FSCAV parameters, was estimated using linear regression in stagnant in vitro solutions (buffer and 75 nM serotonin solutions). An electronic relay was employed to switch between the serotonin-specific waveform described above and a constant potential for a period of controlled adsorption.

### 2.6. Computational Methods

#### 2.6.1. Data Processing

All data processing and analysis was performed using a custom-designed Python 3.6 code. For FSCAV signals, intervals of integration of the faradaic peak were automatically found using a local minima detection algorithm developed in The Analysis Kid [21]. Each interval of integration was visually inspected and corrected when the program failed to detect the faradaic peak of interest. The numerical integral between the faradaic peak and a linear baseline was calculated using the Simpson’s rule. Concentration vs. time traces were calibrated using an electrode-specific linear regression or standardized calibration based on a neural network regression model, which used the whole cyclic voltammogram to obtain a prediction [21]. 

Charge vs. time traces were filtered using a 3rd order band-pass Butterworth filter with a designed passband between 0.0008 Hz and 0.005 Hz. The low cutoff frequency was chosen to remove the low frequency drug effects on the absolute serotonin concentration from consideration. The high cutoff frequency removed the noise component of the signal. Autocorrelation and SWC using Pearson’s correlation coefficient were used to study the relationships between observations in the time series as a function of delay. The autocorrelation function, expressed in Equation (1), was used to calculate the correlation between pairs of samples, distanced by a lag p, where $\bar{X}$ denotes the mean value across the time series of length N [22].
(1)$$r_{p}=\frac{\sum_{i=1}^{N-p}\left(X_{i}-\bar{X}\right)\left(X_{i+p}-\;\bar{X}\right)}{\sum_{i=1}^{N}{\left(X_{i}-\bar{X}\right)}^{2}}$$

Similarly, SWC was used to study the association of fixed-length windows. Equation (2) shows the general formulation of the correlation function between a window of length *l*, starting at time *t,* and another window of the same length, distanced by a lag, *p* [23].
(2)$$r_{t,p}=\frac{\sum_{s=t}^{t+l-1}\left(X_{s}-\bar{X_{s}}\right)\left(X_{s+p}-\bar{X_{s+p}}\right)}{\sqrt{\left(\sum_{s=t}^{t+l-1}{\left(X_{s}-\bar{X_{s}}\right)}^{2}\right)\left(\sum_{s=t}^{t+l-1}{\left(X_{s+p}-\bar{X_{s+p}}\right)}^{2}\right)}}$$

Autocorrelation and SWC were calculated for all delays to study the presence of oscillatory patterns in the time series. Correlation scores were tested for a significant difference from 0 and the highest autocorrelation, obtained from random white noise and the in vitro time series.

White noise was generated by drawing pseudo-randomized samples from a Gaussian distribution, with a mean of 0 and a standard deviation matching the mean standard deviation of all the in vitro experiments presented in this study. The seed of the pseudo-random generator was changed for each white noise trace generated. The PSD of the time series was calculated from the fast Fourier transform spectrum using the Welch method [24]. Analysis of fluctuation patterns pre- and post-drug administration were performed by comparing the PSD peak that corresponded to the largest fluctuation pattern established with the correlation measurements. Additionally, a power-weighted sum of frequencies from the time series spectra was used to evaluate the frequency shift under physiological or pharmacological conditions, following Equation (3).
(3)$$WF=\sum_{f=0}^{f_{max}}P{\left(f\right)}_{norm}\cdot f$$
where *WF* is the power-weighted sum of frequencies, *f_max_* is the maximum frequency of the spectrum, and $P{\left(f\right)}_{norm}$ is the normalized power of a particular frequency, *f*.

#### 2.6.2. Statistical Analyses

Statistical significance is defined as *p* < 0.005. All statistical tests were performed using Python 3.6 SciPy library [25] and Matlab 2020b. Distribution of samples are shown as mean ± standard error of the mean (SEM), unless stated otherwise. Correlation coefficients were tested to be significantly different between groups of signals using a one-way independent ANOVA and Tukey–Kramer *post hoc* multiple comparisons. Statistical significance of changes in the PSD peak and the weighted sum of frequencies were tested using Student’s paired *t*-tests. Full results of the statistical analyses can be provided on request. 

## 3. Results

### 3.1. FSCAV Measurements of Serotonin in an Extended In Vivo Time Window

Figure 1A shows the experimental procedure followed to obtain the FSCV and FSCAV measurements from the CA2 region of a mouse hippocampus, together with a representative acquisition for each of these techniques. The figure includes a sagittal section of a mouse brain, illustrating the positions of the electrodes. The color plots are the raw FSCV and FSCAV data and are described at length elsewhere [26]. Voltage is presented on the *y* axis, time is on the *x* axis, and the current is in false color. We chose to measure in the CA2 region of the hippocampus because of its significance to depression [27,28,29,30]. Additionally, our own previous work has shown that serotonin in this region is very sensitive to paradigms associated with depression (such as chronic stress, inflammation, and neurodegeneration). A 300-min period of serotonin FSCAV measurements, in the absence of an external stimulus, is shown in Figure 1B. A 50 min window is shown directly underneath to illustrate, visually, the presence of oscillations, with time periods of around 10 min. Figure 1C shows the values of correlation, calculated with a SWC (window size 25 min), to calculate the values of correlation across the signal. Each intensity value in the matrix is the Pearson’s correlation coefficient (r) between the 25-min sliding window and another window of the same length. The delay terms determined the first time point of the window in the time series. The diagonal values correspond to correlation values, where the delay of both windows was the same and thus always 1. The correlation matrix presented a pattern of diagonal peaks and valleys, showing that the in vivo signal has periodicity. There are not many external sources of noise with a 10-min time period. Nonetheless, we sought to verify that this periodicity was not associated with signals external to the mouse’s brain.

### 3.2. Comparison of In Vivo and In Vitro Serotonin Recordings

Next, we tested whether any outside sources may have contributed to the fluctuations that were observed *in vivo*. We compared those measurements to identical measurements taken in vitro FSCAV in 75 nM tris-buffered serotonin and randomly generated white noise. 

Figure 2A shows the in vitro FSCAV acquisition of 75 nM serotonin, before and after filtering (light and dark blue, respectively), as well as the calculated autocorrelation and SWC post-filtering. The spike features of the raw time series did not come from changes in the serotonin concentration in the solution, but likely from technical noise and other environmental changes. This results in a low autocorrelation and no significant fluctuation patterns in the SWC analysis. We then compared this to synthetically generated white noise with an average standard deviation taken from a group of in vitro acquisitions (n = 5, 60-min FSCAV acquisitions of 75 nM tris-buffered serotonin, std dev = 0.33) (Figure 2B). The maximum autocorrelation of the in vitro traces was not significantly different from those found in the white noise traces (*post hoc t*-test, r = 0.16 ± 0.04 vs. 0.14 ± 0.03, *p* = 0.9613). In vivo autocorrelation was found to be significantly different from in vitro (*post hoc* independent *t*-test, r = 0.47 ± 0.10 vs. 0.16 ± 0.04, *p* = 0.0093) and white noise (*post hoc* independent *t*-test, r = 0.47 ± 0.10 vs. 0.14 ± 0.03, *p* = 0.0151). 

This finding validates the physiological nature of the oscillatory patterns in the in vivo signals. We then investigated whether there were differences in these patterns between male and female mice, in behavioral paradigms, and after administration of SSRIs.

### 3.3. Physiological, Behavioral, and Pharmacological Serotonin Level Fluctuations

To test whether physiological, behavioral, or pharmacological models affected the frequency of the oscillations, we analyzed the power spectrum density for the range of frequencies found to have a significant correlation using autocorrelation analysis (determined above in Figure 2; 0 Hz to 0.025 Hz). A power spectrum expresses the power (square of amplitude) vs. frequency of oscillations and thus allows for the simple analysis of frequency changes between models. Figure 3A (top) shows the low-pass-filtered raw data, the middle panel shows the average and the SEM normalized PSD, and the bottom panel shows violin plots (distribution of the sum of power-weighted frequencies) for female and male mice (n = 5 animals in each group).

The frequency with maximum power amplitude was not found to be significantly different between male and female mice (independent *t*-test, peak frequency = 0.07 ± 0.01/min vs. 0.08 ± 0.01/min, *p* = 0.2089). The bottom panel shows that there were no statistically significant differences in the power-weighted sum of frequencies between male and female mice (independent *t*-test, WF = 0.18 ± 0.01/min vs. 0.19 ± 0.02/min, *p* = 0.4374). Next, we studied differences in a behavioral model (Figure 3B). Analyzing the same metrics (low-pass-filtered raw data, average and SEM normalized PSD, and violin plots) for animals that had undergone a 16-week, chronic mild stress paradigm vs. the age-matched controls, we again found no significant differences in maximum power amplitude (independent *t*-test, peak frequency = 0.08 ± 0.00/min vs. 0.08 ± 0.01/min, *p* = 0.4422) and sum of power-weighted frequencies (independent *t*-test, WF = 0.18 ± 0.01/ min vs. 0.20 ± 0.02/min, *p* = 0.8180).

There is clear animal-to-animal variability in these (and all) in vivo experiments that might prevent differences from being appreciated. However, we did find differences in one instance where the data was paired and thus variability amongst animals was removed. We observed the oscillations before and after acute administration of escitalopram (n = 7 mice, 10 mg/kg) (Figure 3C). We found that the power of the spectrum trended towards an increase in peak oscillation frequencies following escitalopram administration (paired *t*-test, peak frequency = 0.07 ± 0.00/min vs. 0.08 ± 0.01/min, *p* = 0.2031). We also observed a significant change in the power-weighted sum of frequencies in the control group after administration of escitalopram (paired *t*-test, WF = 0.28 ± 0.02/min vs. 0.34 ± 0.02/min, *p* = 0.0285). 

A limitation of this analysis is the acquisition frequency of FSCAV (1 sample per minute); to be conservative (i.e., certain that we were not including noise in the analysis), we did not consider frequencies higher than 0.005 Hz. Therefore, we are unable to say whether there was important physiological information at higher frequencies that were not captured because of the method’s current configuration.

### 3.4. FSCAV with Higher Temporal Resolution

The initial development of FSCAV comprised a background time (2 s), adsorption time (10 s), redox time (18 s), and waiting time (30 s) (see Materials and Methods) that enabled a 1-min temporal resolution. Figure 4A shows a representative example of an in vivo FSCAV color plot, wherein each of the acquisition steps are marked. Here, these parameters were altered to determine whether temporal resolution could be improved. Figure 4B shows the ratiometric change in sensitivity from the control in vitro acquisition (n = 3 electrodes, 30 files per acquisition) when one or more parameters were changed (see figure caption). Using a repeated measures ANOVA, we found that a decrease of the wait time from 30 s to 0 s did not significantly change the sensitivity of FSCAV for serotonin detection (*post hoc* paired *t*-test, sensitivity = 47.05 ± 10.23 pC/µM vs. 40.42 ± 9.37 pC/µM, *p* = 0.7503). Additionally, changing the background time from 2 s to 1 s did not significantly change the sensitivity (*post hoc* paired *t*-test, sensitivity = 47.05 ± 10.23 pC/µM vs. 40.07 ± 12.67 pC/µM, *p* = 0.6792). As expected, a strong decrease in sensitivity was observed when the adsorption time was changed from 10 s to 1 s (*post hoc* paired *t*-test, sensitivity = 47.05 ± 10.23 pC/µM vs. 4.14 ± 0.69 pC/µM, *p* = 0.0174). Changing the redox time on top of this did not have a significant effect on the sensitivity (*post hoc* paired *t*-test, sensitivity = 4.14 ± 0.69 pC/µM vs. 4.13 ± 0.61 pC/µM, *p* = 0.4198). The faradaic peak remained detectable and quantifiable despite the decrease in sensitivity, as depicted in Figure 4C, which shows a representative FSCAV color plot with the new experimental procedure. The total acquisition time was reduced from 60 s to 14 s. We then set out to apply this new acquisition paradigm *in vivo*.

Figure 4C shows a representative experiment with conventional FSCAV for detecting serotonin in the CA2 region of the hippocampus pre- and post-escitalopram administration (10 mg/kg). Figure 4D shows the PSD analysis pre- and post-drug. Figure 4E,F show the same analysis using the fast FSCAV time resolution. Here, escitalopram increased the steady-state levels of serotonin from 34.20 ± 0.26 nM (first n = 300 samples) to 41.24 ± 3.15 nM (last n = 5 samples) 70 min after injection. As shown in Figure 4D,E (with the slow time resolution), there was a shift of the PSD to higher frequencies, as evidenced by the measured power-weighted frequency (pre-escitalopram = 0.16/min, post-escitalopram = 0.18/min). In addition, the higher temporal resolution allowed us to observe higher frequency components, along with a decrease in oscillation amplitude after drug administration (pre-escitalopram = 20.21 nM^2^, post-escitalopram = 13.26 nM^2^). This change in amplitude was not seen in the conventional FSCAV acquisition (pre-escitalopram = 8.37 nM^2^, post-escitalopram = 8.19 nM^2^).

## 4. Discussion

### 4.1. FSCAV Reveals Ambient Serotonin Fluctuations In Vivo

The conventional gold standard technique for measuring basal or ambient neurotransmitter concentrations is microdialysis, commonly coupled with high-performance liquid chromatography and mass spectrometry. These approaches have excellent sensitivity and selectivity, and there have been recent improvements in acquisition frequency [31] and the size of the probes [32], which have helped to solve previous issues related to the slow temporal resolution of data acquisitions and tissue damage/inflammation.

Electrochemical methods have enabled faster, more localized serotonin measurements by using FSCV at CFMs, with negligible tissue damage [33,34]. A major disadvantage of FSCV was its limited ability to only report changes in concentration and not basal levels. In response to this limitation, FSCAV was developed [10], which we then applied to in vivo serotonin measurements [12]. We used this method to study basal serotonin levels in different brain regions, between male and female mice, and in response to different pharmacology [5,8,13].

In this paper, we observed an interesting phenomenon after analyzing raw FSCAV serotonin data that were collected from the mouse hippocampus in vivo over a period of several hours, in the absence of stimuli. We observed sustained fluctuations around a mean value, with an approximate peak-to-peak concentration amplitude of 10 nM and a periodicity of 10 min, which did not occur *in vitro*. The observation of oscillatory neuronal activity has allowed for the application of network theory and plausible mechanisms for the storage, co-ordination, modulation, and transfer of information across the brain [35]. Brain waves impose a spatiotemporal structure on neural connectivity within and across brain areas, which has been shown to be essential for cognitive processes such as attention and memory [36]. Power spectrum analysis reveals a wide range of neuronal oscillations within specific frequency bands, ranging from <0.01 Hz to >1,000 Hz [37], with fast waves typically localized to restricted neural volumes, and slower oscillations driven by synchronous membrane voltage fluctuations across wider brain areas [38]. Crucially, these neuronal oscillations interact across different frequency bands to modulate each other and engage in specific behaviors [39]. Moreover, oscillation phase relationships, defining connectivity across brain regions, can be modulated by sensory inputs [40]. 

The oscillatory activity that we describe here is unusual, in that the frequency is much lower than even the slowest brain waves typically described in the sleeping brain. The vast majority of literature referring to the mechanisms of oscillatory neuronal activity has focused on factors within the central nervous system itself, including variations in the concentration of extracellular neurotransmitters and ions, intrinsic circuitry, speed of axonal conduction, and changes in cellular excitability. However, the brain is also intricately linked to the peripheral nervous system and a plethora of endocrine cues. Indeed, many peptides that systemically act as classical hormones are also produced locally as neurotransmitters and neuromodulators within the brain. Neuroendocrine modulation of brain activity by peripheral hormones, that either cross the blood brain barrier, interact directly with parts of the brainstem and hypothalamus, or that modulate peripheral sensory inputs, is well described, particularly in the fields of body weight regulation and metabolism [41]. Almost all of these hormones are released in a pulsatile fashion, the teleological explanation for which is that pulsatility allows for their receptors to reset, recycle, and avoid downregulation.

Here, we propose the tantalizing concept that peripheral hormone cyclicity may be an important component of neuronal modulation and neuroendocrine connectivity. For example, insulin pulsatility in the blood occurs at the same periodicity as the serotoninergic waves that we have measured in this work [42]. There is wide evidence that serotonin and insulin are co-secreted in the pancreas [43], and that bodily insulin readily transfers to the brain via a saturable transporter [44,45]. Insulin signaling is important in the hippocampus [46], and there is mounting evidence that insulin modulates serotonin, providing the basis for a link between mood and metabolic state [46,47]. Aside from the complex bidirectional relationship between diabetes and mood in humans, animals with both type 1 and 2 diabetes also display depressive-like phenotypes, suggesting a biological link that is separate from the obvious psychosocial burden of a chronic disease [48,49]. The depressive-like behavior in animal models was seen to improve after the stabilization of blood insulin [50,51]. This finding strengthens the connection between serotonin and mood since the hallmark of type 2 diabetes is the early loss of pulsatile insulin secretion—in fact, some have argued that this is the primary physiological insult [52]. Thus, the ability of FSCAV to measure serotonin oscillations over physiologically relevant time frames could open the door to a more detailed understanding of this important missing piece of the puzzle of brain connectivity.

### 4.2. Serotonin Extracellular Fluctuations Modeled by Serotonin Transporter Modulation

To further understand the nature of these oscillations, we first compared the cycling patterns in ambient hippocampal serotonin between male and female mice. There is conflicting evidence in the literature about differences in the serotonin systems between sexes. In our own previous work, we found no differences in control serotonin release and reuptake, nor in the basal levels, between male and female mice with FSCV and FSCAV [8]. In the present study, we also found no significant differences in oscillations, which is consistent with our previous findings. Next, we turned to prior work that had unearthed differences in ambient serotonin levels between models. First, in the prior work, we found a robust reduction in ambient hippocampal serotonin levels in mice that were exposed to a CMS paradigm, regardless of their behavioral phenotypes [14]. Second, since we have shown many times that acute SSRI administration increases ambient serotonin levels [8,13], we analyzed the oscillations in these two sets of data. We found no differences between animals that were exposed to chronic stress vs. their age-matched controls. In our prior work, we showed that, functionally, there were no differences in how serotonin was released and reuptaken (*via* no differences in evoked serotonin release) between control and CMS mice. Rather, the lower ambient levels in CMS mice were hypothesized to result from inhibition from another modulator. Thus, this shows that so long as the release and reuptake machinery is intact, the oscillation frequency does not change. Indeed, we found that after active uptake inhibition with escitalopram, the power spectrum shifted to a higher band.

Our hypothesis for the control of these oscillations is found in Figure 5 below. The upward cycle in (C) shows serotonin progressively accumulating in the extracellular space. This could be due to constant activation of serotonin release by a modulator whose concentration is also progressively increasing [47]. Therefore, at this stage, the production of serotonin release outcompetes reuptake. At a certain threshold concentration, serotonin begins to fall, showing that reuptake mechanisms have been triggered, such that they outcompete the release mechanisms. It is likely that this effect is autoreceptor-mediated, as there is good evidence that 5-HT_1B_ autoreceptors directly modulate serotonin transporter (SERT) activity [53,54,55]. Once a lower threshold concentration is reached, the release mechanisms outcompete the reuptake mechanisms—again, likely autoreceptor-mediated, since 5-HT_1A_ autoreceptors modulate serotonin release [56,57,58]—and serotonin begins to rise.

Adding nuance to the story is that serotonin is reuptaken via two transport systems [4,6,59], uptake-1 and uptake-2. Uptake-1 are the SERTs, and represent a high-affinity, low-capacity reuptake system. Uptake-2 is a combination of dopamine transporters (DATs), norepinephrine transporters (NETs), and organic cation transporters (OCTs) and is a low-affinity, high-capacity system. When SERTs are blocked, the high-capacity uptake-2 system becomes more active, thus the speed of cycling should be faster than in a SERT-only controlled system (*i.e.,* lower capacity). Indeed, we found that, after escitalopram, the frequency of the cycles increased. Although there are many drugs with high potency for SERTs [60,61], we chose to utilize escitalopram because of its high selectivity for SERTs.

While it is very likely that other aspects of the serotonin signaling machinery, such as the autoreceptors, are involved in regulating the oscillations, it is clear that transporters control the oscillation frequency.

### 4.3. Fast FSCAV Improves Spectrum Resolution of Serotonin Fluctuations

While FSCAV’s one sample per minute is highly informative, with this sampling frequency there is a chance that the spectrum of analyzed frequencies suffers from aliasing because the physiological signal we seek may contain higher frequencies. We thus increased the acquisition frequency of FSCAV. We found, that although the sensitivity strongly decreased as adsorption time was reduced, we could still capture in vivo signals. With this much faster acquisition (14 s) we measured in vivo data from the CA2 region of the mouse hippocampus. The dominant oscillation frequency at this faster acquisition rate was around 6.5 min (compared to around 10 min at the slower acquisition rate). We found this result interesting, again, in the context of endocrine signaling. Early studies reported that insulin cycled in vivo with a period of approximately 10 min, but as researchers developed better and faster tools to measure insulin, the oscillations were reported as being closer to 5 min [62]. Both observations are classic examples of signal aliasing—where signals are distorted due to a low sampling frequency [63].

We found that, after escitalopram, and using this faster acquisition rate, there was a more substantial shift of the power spectra to higher frequencies—again speaking to the uptake-2 processes being activated. We also observed a visible change in the amplitude of the fluctuations after administering the drug, perhaps as a result of autoreceptors trying to control the release after the SSRI starts to increase the overall basal concentration of serotonin. It has long been known that SSRIs increase somatodendritic serotonin release in the cell bodies that inhibit cell firing via 5-HT_1A_ autoreceptors [64].

Therefore, increasing the temporal resolution of the method provides much richer and more accurate physiological information.

## 5. Conclusions

FSCAV measures minute-to-minute changes in ambient extracellular serotonin, which we have used in the past to find critical information about brain physiology and disease. Here, we carefully analyzed in vivo FSCAV data and found strong oscillations in the basal serotonin levels, which had a 10-min period and were not present *in vitro*. These oscillations did not differ between male and female mice, nor did they differ between mice that underwent a chronic stress paradigm and those in the control group. Acute SSRI administration caused a shift in the oscillation frequency, leading us to hypothesize that the fluctuations were transporter mediated. We optimized the experimental FSCAV parameters to measure at a higher temporal resolution and observed more pronounced shifts in the oscillation frequency, and a decreased oscillation amplitude. We suggest that this work may potentially serve as a new model for investigating the interactions of serotonin with endocrine messengers that fluctuate on the same time scale.

## Figures and Tables

**Figure 1 cells-11-01719-f001:**
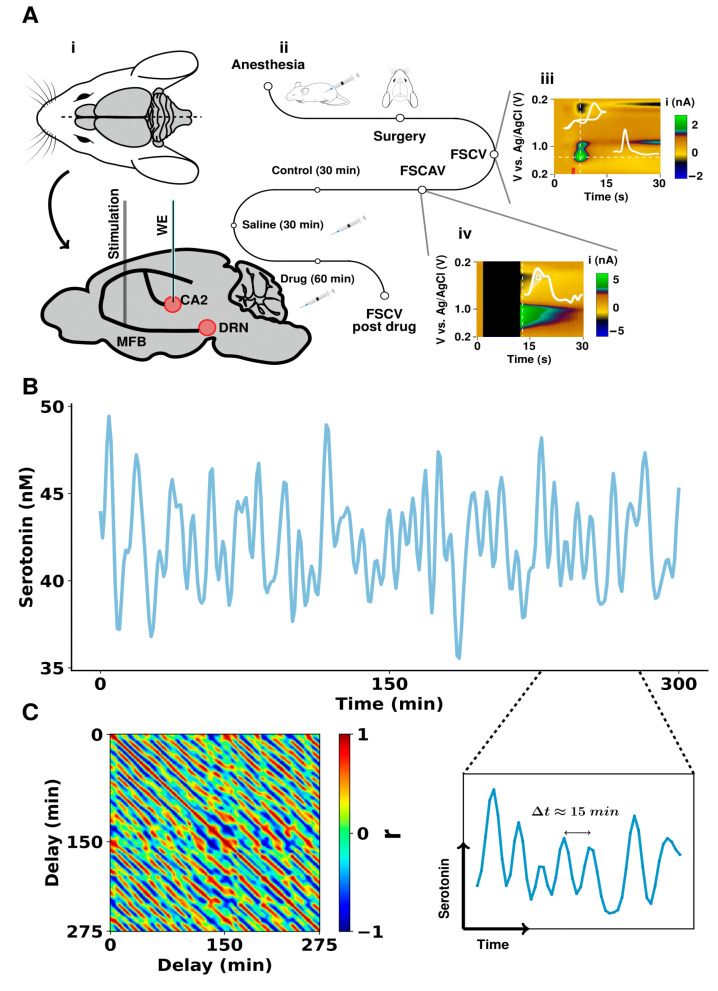
In vivo hippocampal serotonin fluctuations in mice. (**A**) Schematic of experimental paradigm for serotonin measurements with FSCV (evoked) and FSCAV (basal). (**Ai**) shows the position of the stimulation electrode in the medial forebrain bundle and the working electrode (WE) in the CA2 region of the hippocampus. The reference electrode was positioned in the opposite hemisphere of the mouse brain. (**Aii**) shows the timeline for experiments where an antidepressant was acutely administered to the mouse. (**Aiii**,**Aiv**) show a raw FSCV acquisition and an FSCAV acquisition, respectively, with inset cyclic voltammogram and a current vs. time trace. The marked area under the curve (charge, Q) was used to obtain estimations of basal serotonin levels. (**B**) A 300-min in vivo FSCAV acquisition from the CA2 region of a mouse hippocampus. A 50-min section of this is also directly shown below. The signal was low-pass filtered with a 3rd order Butterworth filter and with a 0.005 Hz cutoff frequency. (**C**) Sliding window correlation of the time series (shown in (**A**)) with itself, with a window size of 25 min. The high (positive and negative are presented in red and blue, respectively) correlation stripes show the repetitive fluctuation pattern of the time series.

**Figure 2 cells-11-01719-f002:**
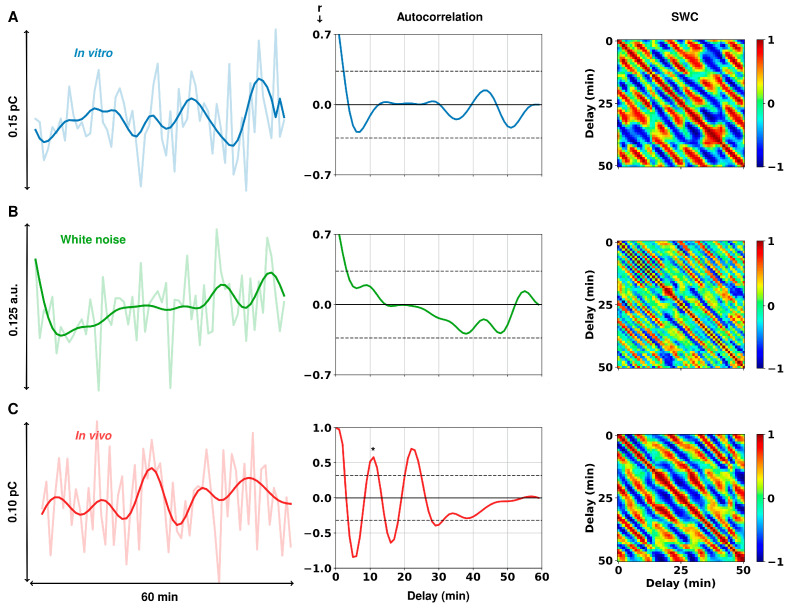
Representative FSCAV autocorrelation analysis. (**A***)* In vitro FSCAV serotonin acquisition in 75 nM tris-buffered serotonin, shown in charge units (pC = picocoulombs); (**B**) random white noise with a standard deviation equal to the in vitro acquisition; and (**C**) in vivo FSCAV serotonin acquisition in the CA2 region of a mouse hippocampus, shown in charge units. The graphs on the left show both the unfiltered and filtered signals (3rd order Butterworth low-pass filter with 0.005 Hz cutoff frequency). The graphs in the center show the autocorrelation of each respective time series after filtering. Gray dashed lines show the limits of statistical significance. On the right are shown the SWC, as a function of the delay between 10-min windows of the respective time series.

**Figure 3 cells-11-01719-f003:**
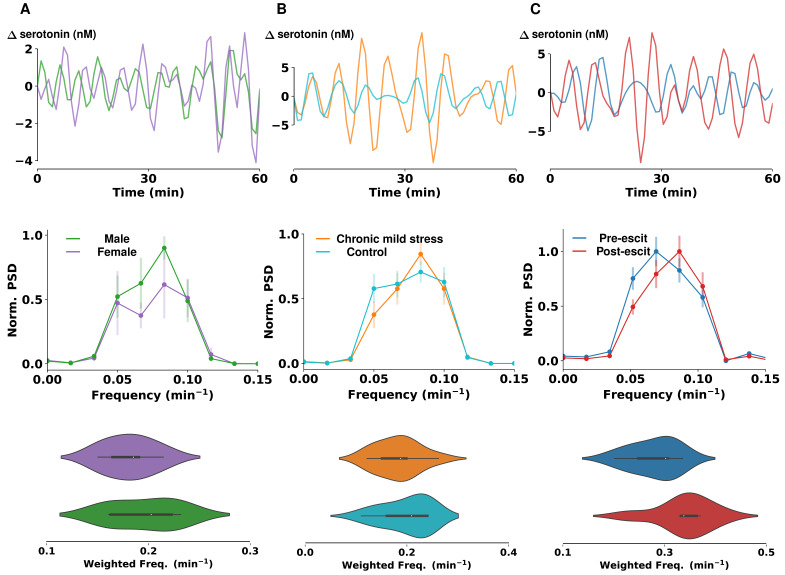
Spectral analysis and comparison of serotonin oscillations. (**A–C**) Representative examples of 60-min FSCAV serotonin acquisitions in the CA2 region of a mouse hippocampus. Each signal was band-pass filtered with a 3^rd^ order Butterworth filter, with cutoff frequencies of 0.0008 Hz and 0.005 Hz. The power spectrum of each acquisition was then computed and the average and SEM across animals is represented in the center panels. (**A**) shows differences in the PSD of male and female mice (n = 5 animals in each group); (**B**) shows differences between mice that underwent a chronic mild stress paradigm and the control mice (n = 10 animals in each group); and (**C**) shows differences in PSD, pre- and post-escitalopram (escit) (10 mg/kg) administration, in mice (n = 5 animals). Power-weighted sum of frequencies were calculated for each power spectrum, and violin plots of their distributions for each dataset are shown below.

**Figure 4 cells-11-01719-f004:**
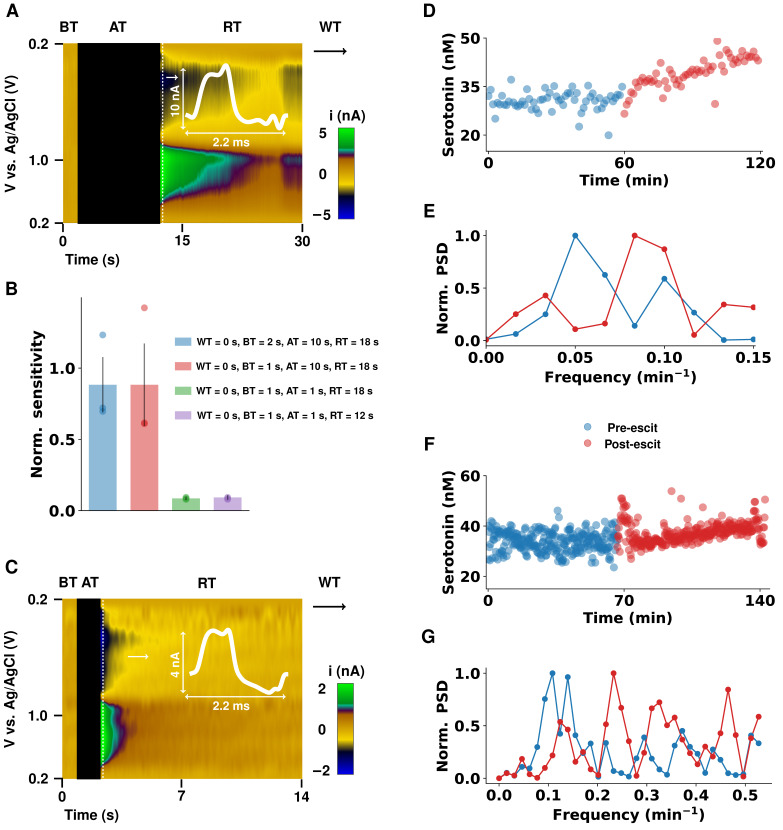
Comparison of conventional FSCAV with fast FSCAV acquisition (14 s sampling period). (**A**) Representative example of in vivo FSCAV color plot from the CA2 region of a mouse hippocampus, using the conventional FSCAV method with inset cyclic voltammogram. Labels of the different steps of the FSCAV acquisition are as follows: background time (BT), adsorption time (AT), redox time (RT), and wait time (WT); (**B**) the ratio of sensitivity between conventional FSCAV (depicted in (A)) and fast FSCAV, with parameters given in the legend (n = 3 electrodes); (**C**) a representative example of in vivo FSCAV color plot from the CA2 region of a mouse hippocampus, using the fast FSCAV method (WT = 0 s, BT = 1 s, AT = 1 s, and RT = 12 s) with inset cyclic voltammogram; (**D**) concentration vs. time from an FSCAV acquisition, drawn from the CA2 region of the hippocampus of an anesthetized mouse, pre- and post-escitalopram administration (10 mg/kg), using the conventional FSCAV method; (**E**) a normalized power spectra density (PSD) for the representative acquisition given in (**D**); (**F**) concentration vs. time from an FSCAV acquisition drawn from the CA2 region of the hippocampus of an anesthetized mouse, pre- and post-escitalopram administration (10 mg/kg), using the fast FSCAV method shown in (**C**); and (**G**) a normalized power spectra density (PSD) for the representative acquisition given in (**F**). The time series were band-pass filtered to remove the slow-frequency effects of escitalopram and the high-frequency noise. The concentration vs. time trace shown in (**D**) was filtered following the filtering paradigm described in Materials and Methods. For the time series in (**F**), the high cutoff frequency was, in this case, set to 0.02 Hz (60% of Nyquist frequency), while the lower cutoff frequency was maintained at 0.0008 Hz. The higher acquisition frequency allows the study of higher spectrum frequencies.

**Figure 5 cells-11-01719-f005:**
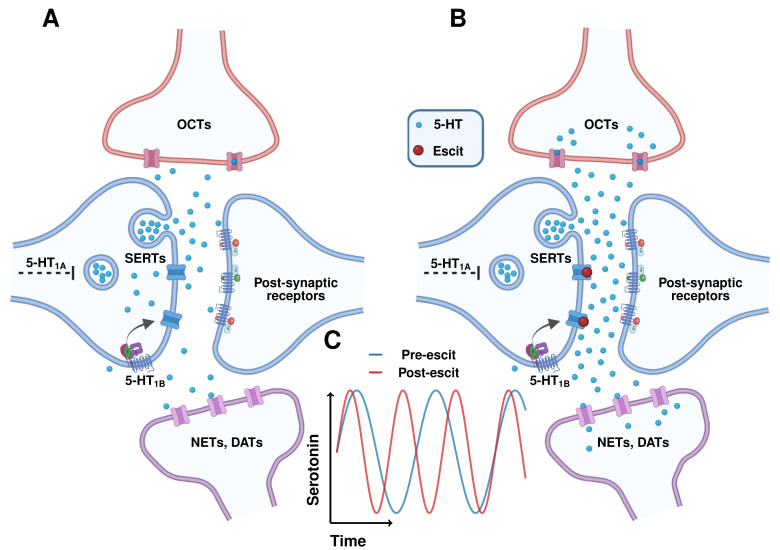
Illustration of control of oscillations. (**A**) Schematic of synaptic elements involved in serotonin release and reuptake in the control state. Serotonin is released from the presynaptic neuron and uptaken, both by local serotonin-specific transporters (SERTs; uptake-1) and other monoamine transporters present in glia and adjacent neurons (NETs, DATs, and OCTs; uptake-2). (**B**) Schematic of synaptic elements involved in release and reuptake in the presence of escitalopram. Here, serotonin is mostly reuptaken via the uptake-2 mechanisms, which have a low affinity for serotonin, but have a high capacity due to the large number of transporters. (**C**) Illustrative representation of extracellular serotonin fluctuations and their increase in frequency in the presence of escitalopram due to the activation of the high-capacity uptake-2 system. Created with Biorender.com (accessed on 6 May 2022).

## Data Availability

Data are available upon request.

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
