# Peer review of "Low-Frequency Oscillations of In Vivo Ambient Extracellular Brain Serotonin"

_cells, 2022, doi:10.3390/cells11101719_

Round 1
Reviewer 1 Report
Summary:
The manuscript by Witt et al. is of interest to researchers that investigate or want to investigate neurotransmitter dynamics within specific regions of the brain. In this manuscript, the authors employ in vivo fast-scan adsorption-controlled voltammetry (FSCAV) to measure serotonin (5-HT) level dynamics in mouse hippocampus. On further optimization of experimental protocols and analyses employing statistical correlation, the authors record strong basal oscillation in 5-HT levels over several hours. Furthermore, they report shift in oscillation frequencies on acute administration of an anti-depressant with no frequency changes in control male and female mice and in mice that are subjected to a chronic stress behavioural paradigm. This is an interesting study that provide potential insights into cross-talk of peripheral and endocrinal signals with neuromodulation at the central nervous system under physiological and pathological conditions.
The manuscript is generally very well-written with rigorous data analyses but the figures lack raw traces. The inclusion of additional figures that illustrate the methodology of FSCAV and raw traces of the experiments undertaken in each figure (see comments and recommendations) can make the manuscript an easier read to a general reader along with making the figures bulkier. In addition, providing justification for certain aspects of experimental methodology will make the paper more accessible to a wider audience. Throughout the whole manuscript, statistical details in comparisons are missing in the figure legends, which have to be addressed.
Following are few comments and recommendations for the authors to address:
- The voltage range specifically applied to measure serotonin levels using FSCAV should be mentioned in the methods. This information would help the readers understand the technique better for its sensitivity and specificity to 5-HT detection.
- Escitalopram was administered acutely and the recordings were done assumedly on the same day after SSRI injection. Apart from attributing the experimental observations to uptake-1 and 2 systems, what is the justification of looking at acute effects of SSRI on brain 5-HT dynamics when conventional knowledge states that onset of action of SSRIs are typically 4-6 weeks? The FSCAV recordings may reveal more information of 5-HT oscillations if SSRIs are administered in normal dosage daily as opposed to acutely. For acute effects on serotonin fluxes, were SERT-preferring amphetamines (such as pCA/fenfluramine) considered? This should be explained briefly either in the materials and methods or discussion sections.
- Serotonin fluxes occur in different regions of the brain. The working electrode was introduced in the CA2 region of the mice hippocampus. Why this region over medial pre-frontal cortex, hypothalamus, nucleus accumbens or the amygdala that also show serotonin signalling? The authors should add this reasoning in the results or discussion. A cartoon showing the regions where the three electrodes have been introduced in the mouse brain should be added in Fig.1.
- The calibration curves by which 5-HT concentrations were determined on the basis of charge changes during redox is not clearly explained in the material and methods and should be explained better. An example voltammogram (with raw data, active area, deconvolved data) of FSCAV acquisition over the 60s period and voltage-current plots should be included in Fig.1.
- In Fig.2, two of the acquisitions have pC (picocoulomb?) and the one by random white noise has a.u. as units. Is this because the latter was generated as opposed to recorded? In the figure legend, the following statement is written “Random white noise with standard deviation equal to the in vivo acquisition”. It should be “in vitro acquisition”.
- The authors should explain the chronic mild stress behavioural paradigm in the methods section or at least provide a reference where it is stated.
- In the middle panels of Fig.3, normalised power spectrum densities of pre-escit mice, which is ideally control animals, should match that of control mice in the stress paradigm experiments and that of means of male and female control mice. But the distribution profiles of frequencies look quite different. Can the authors explain these differences? Were stressed mice treated with escitalopram and their PSDs compared to control and control + escitalopram mice? In addition, were there any differences in the escitalopram treated male and female mice? Is p.d.u. power density units?
- Was the shortened FSCAV protocol also done for male vs female mice, control mice vs stressed mice? If yes, were there any observable differences? In Fig.4a, it is recommended to show the raw data of the reduced faradaic current due to the shortened protocol in comparison to the same in the normal protocol. In Fig.4B, box plot indicating the 1 min (BT=2s, AT=10s, RT=18s, WT=30s) is missing. The box plots, I would assume, are normalized to the 1 min protocol because the plot values (unit of y axis) do not match the ones mentioned in the results section. And is it pC/μM (mentioned in the text) or pC/nM (mentioned in the figure)? In the figure legend, the authors should include the (BT=w s, AT=x s, RT=y s, WT=z s) format for all conditions.
- In Fig.4c and 4d, displaying representative traces and frequency plots for both short and long protocols is recommended to show there is an increased frequency of oscillation despite reduced sensitivity in the short protocol. In addition, Fig.4d can have two frequency plots, one comparing long and short protocol with untreated mice and the same for treated mice. This will make it easier to observe the shifts in peak frequencies and amplitudes.
- In Fig.5, the figure legend lacks information. Writing, in short, the proposed differences between the left (a) and right panel (b) (as mentioned in the discussion) that led to increased frequency oscillations after escitalopram treatment is recommended.
- Did the authors consider pairing in vivo serotonin and insulin voltammetric measurements (if possible), like they did in another pre-print for serotonin and histamine? The constant co-mentioning of serotonin and insulin in the manuscript discussion section warrants this investigation.
Other minor comments involving typos are as follows:
- Line 44-45 in the draft, “to measure slower” as opposed to “to measuring slower”.
- Line 167, there is a mention of supplementary information when there is none.
- Line 171, “is shown in Figure. 1A” as opposed to “is show…”.
- Line 327, it should be written as “to be certain that we are not…”.
- Line 415, “plethora of…” as opposed to “plethora….”
Reviewer 2 Report
The manuscript characterizes low frequency oscilations of serotonin level in the brain of the mice that have been measured using fast scan cyclic voltammetry. The work is well designed, documented and comprehensively discussed and is of wide interest for the readers of the Journal. This reviewer does not hold further suggestions to the authors.
Author Response
Thank you for your kind comments about the work presented in the manuscript.
Reviewer 3 Report
In the present manuscript authors use fast-scan controlled-adsorption voltammetry (FSCAV) to show that the basal serotonin levels in the mouse hippocampus oscillates with a 10 min period and the oscillation properties can be modulated by serotonin transporter availability. Authors have previously used FSCAV to measure basal serotonin levels and in the present manuscript authors use the method to explore the oscillatory nature of serotonin in adult mouse hippocampus. Authors use well accepted autocorrelation and sliding window correlation analysis to show a periodicity in serotonergic waves which is visible only in vivo. Authors then go on to compare if there is any effect of gender, stress and SSRI drug treatment. Authors also amend the FSCAV parameters to measure at a higher temporal resolution to show that SSRI treatment shifts the oscillation frequency and decreases the oscillation amplitude.
The manuscript is well written and provides sufficient details for a broader audience.
I do not have any major concerns. However, I would insist authors to thoroughly check for typos across the manuscript. For instance, line 299, 544. Authors should correct the Figure legend 2B, line 258, ´standard deviation equal to the in vivo or in vitro acquisition’.
